# Screening of aspiration pneumonia using the modified Mallampati classification tool in older adults

**Jianping Liu, Hironobu Fukuda** ⬛ *, **Eiji Kondo, Yuki Sakai, Hironori Sakai, Hiroshi Kurita**

Department of Dentistry and Oral Surgery, Shinshu University School of Medicine, Matsumoto, Japan

* hironobu_fukuda@shinshu-u.ac.jp

**Data Availability Statement:** Data cannot be shared publicly because of the risk violating the patients' privacy. All researchers may contact the authors of this study via Ethics Committee of

## Abstract

Pneumonia is a major cause of morbidity and mortality in older adults. In the aging society, screening methods for predicting aspiration pneumonia are crucial for its prevention. Changes in the oropharyngeal morphology and hyoid bone position may increase the risk of aspiration pneumonia. This multicenter study aimed to investigate a simple and effective screening method for predicting dysphagia and aspiration pneumonia. Overall, 191 older adults (aged 65 years or older) were randomly sampled using the simple random sampling technique. Oropharyngeal morphology was assessed using the modified Mallampati classification, which reflects the size of the tongue in the oropharyngeal cavity. The hyoid position was measured as the distance between the menton and laryngeal prominence to evaluate aging-related changes in the muscles of the laryngopharynx. Dysphagia was assessed using the repetitive saliva swallowing test (RSST), which measures the number of swallowing movements in 30 seconds; dysphasia is defined as less than 3 swallowing movements in 30 seconds. The aspiration signs were assessed based on history of choking or coughing reflex during eating or drinking and medical history of pneumonia. The study findings revealed that the modified Mallampati classification was significantly correlated with a medical history of pneumonia. A higher incidence of pneumonia was evident in the lower Mallampati classification, which shows the smaller size of the tongue base in the oropharyngeal cavity. The results of this study suggest that the modified Mallampati classification may be a possible screening method to predict the occurrence of pneumonia.

## Introduction

The health status of the aging population currently poses a global concern. Japan is one of the leading countries with a rapidly aging population [1]. With the gradual rise in the older population, people are demanding a longer healthy life. Aspiration pneumonia ranks sixth among the leading causes of death, accounting for approximately 3% of all deaths according to a demographic survey conducted in Japan in 2019 [2].

Aspiration pneumonia is believed to be associated with oral and swallowing dysfunctions in the older population [3] Currently, swallowing function is clinically assessed by

Shinshu university(contact via mdrinri@shinshu-u.ac.jp), and the data will be available to researchers who meet the criteria, after consultation with the Ethics Committee of Shinshu University.

**Funding:** The author(s) received no specific funding for this work.

**Competing interests:** The authors have declared that no competing interests exist.

videofluoroscopic and videoendoscopic examinations in Japan; however, these examinations are difficult to perform because they require specialized equipment. Meanwhile, the repetitive saliva swallowing test (RSST), water swallowing test, and cervical auscultation of the swallowing test are frequently conducted as convenient screening tools for swallowing function [4]; however, these examinations are time-consuming and pose a risk of aspiration. Thus, there is a need to develop simple and safe examination methods that can be performed by non-experts and without the need for specialized equipment to screen for potential dysphagia.

In daily clinical practice in dentistry, aging-associated changes in the tongue and suprahyoid muscles of older adults are often observed [5]. Swallowing movements are mainly performed by the tongue and suprahyoid muscles; the dysfunction of these muscles is closely associated with dysphagia [6]. Therefore, the evaluation of changes in the oropharyngeal morphology and hyoid bone position may lead to the evaluation of swallowing function, which in turn determines the risk of aspiration pneumonia [3].

There is an urgent need to manage aspiration pneumonia in the aging population. The purpose of this study was to investigate the association between the risk of aspiration pneumonia and the oropharyngeal morphological changes or hyoid bone position.

## Materials and methods

### Research area, settings, period, and sampling

The present study is a clinical trial on dysphasia in randomly selected patients who met the inclusion criteria. All surveys for research purposes were conducted between May 16, 2021, and June 15, 2022. The authors had access to information that could identify the individual participants during or after data collection.

### Participants

In total, 191 patients aged 65 years or older who visited the dental and oral surgery departments at 11 hospitals in Nagano Prefecture and met the following criteria were included in this study. The inclusion criteria were patients who (1) were 65 years or older; (2) had an Eastern Cooperative Oncology Group (ECOG) Performance Status (PS) of 0–3; and (3) agreed to participate in this study. The exclusion criteria were patients (1) with head and neck disease that may impair the swallowing function; (2) with brain and nervous system disorders that may impair swallowing function (e.g., cerebrovascular or cranial nerve disease); and (3) who administered psychoactive medications. The clinical data, including age, sex, height, weight, body mass index (BMI), ECOG PS, past medical history, and medication history, of the patients were retrospectively collected.

### Assessment of swallowing function and signs of aspiration

Swallowing function was assessed using the RSST [7]. In RSST, patients sitting on chairs are instructed to swallow saliva as many times as possible in a time frame of 30 seconds. The examiner counts the number of swallowing movements by lightly placing their fingers on the patient's laryngeal prominence and hyoid bone. This method was used because the number of swallowing movements decrease in patients with dysphasia.

In addition, signs of aspiration were determined as follows. The participants were asked if they had experienced "choking or coughing reflex" from food or drinks in the previous month and if they had a history of pneumonia in the previous year.

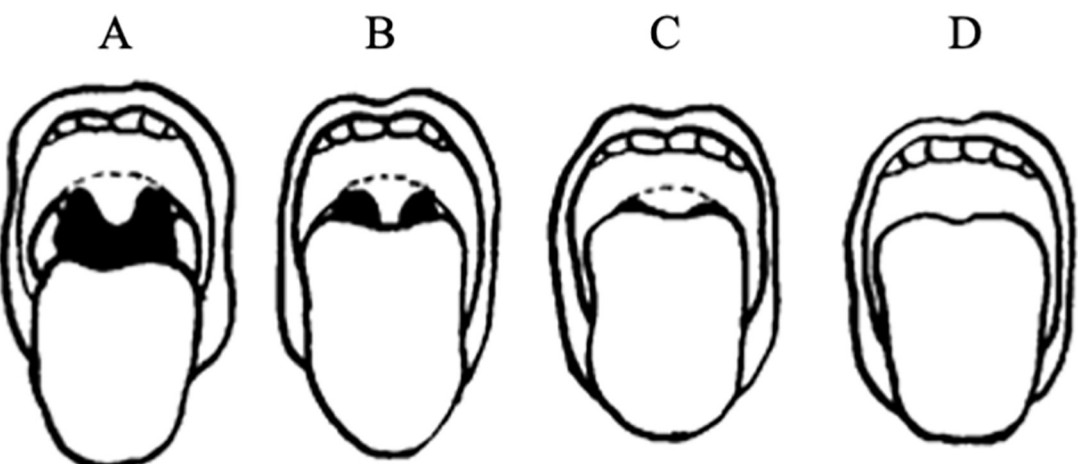

**Fig 1. Modified Mallampati classification (Samsoon GL and Young JR: Difficult tracheal intubation: A retrospective study. Anaesthesia 1987;42:487–490).** (A) Class I. Soft palate, uvula, fauces, and pillars are visible. (B) Class II. Soft palate, uvula, and fauces are visible. (C) Class III. Soft palate and base of uvula are visible.(D) Class IV. Soft palate is not visible at all.

## Evaluation of the oropharyngeal morphology (modified Mallampati classification)

The Mallampati classification was developed to predict the degree of intubation difficulty [8]. In recent decades, Samsoon and Young's modified Mallampati classification has been widely used in anesthesiology [9]. In this study, the modified Mallampati classification was used to evaluate the oropharyngeal morphology (Fig 1). In Mallampati class I, the soft palate, uvula, fauces, and pillars are visible. In Mallampati class II, the soft palate, uvula, and fauces are visible. In Mallampati class III, the soft palate and base of the uvula are visible. In Mallampati class IV, the soft palate is not visible at all. To standardize the conditions in this investigation, the patients sat upright and their heads were kept in a neutral position by all the evaluators. They were asked to protrude their tongues and keep their mouth open. The evaluators received practical training on the modified Mallampati classification to ensure uniformity in the conducted evaluation.

## Evaluation of hyoid bone position (distance from the menton to the laryngeal prominence)

The hyoid bone hangs from the mandible by the suprahyoid muscles. The position of the hyoid bone relative to the mandible was measured. In previous studies investigating aging-related changes in the laryngopharynx, the position of the hyoid bone was often used [10]. As it is often difficult to find the position of the hyoid bone, we used the easy-to-understand laryngeal prominence as a surrogate marker. We measured the linear distance between the mandibular menton and laryngeal prominence as an indicator of hyoid bone position (Fig 2). To define the regular positions, the participants were kept facing forward in seated postures and instructed to relax their neck muscles. Measurements were taken in millimeters and evaluated as rank data for each centimeter.

## Statistical analyses

All data were compiled and analyzed using SPSS Statistics (version 27; IBM Corp., Armonk, NY, USA). To investigate the correlation of characteristics, Student's *t*-test and Mann–

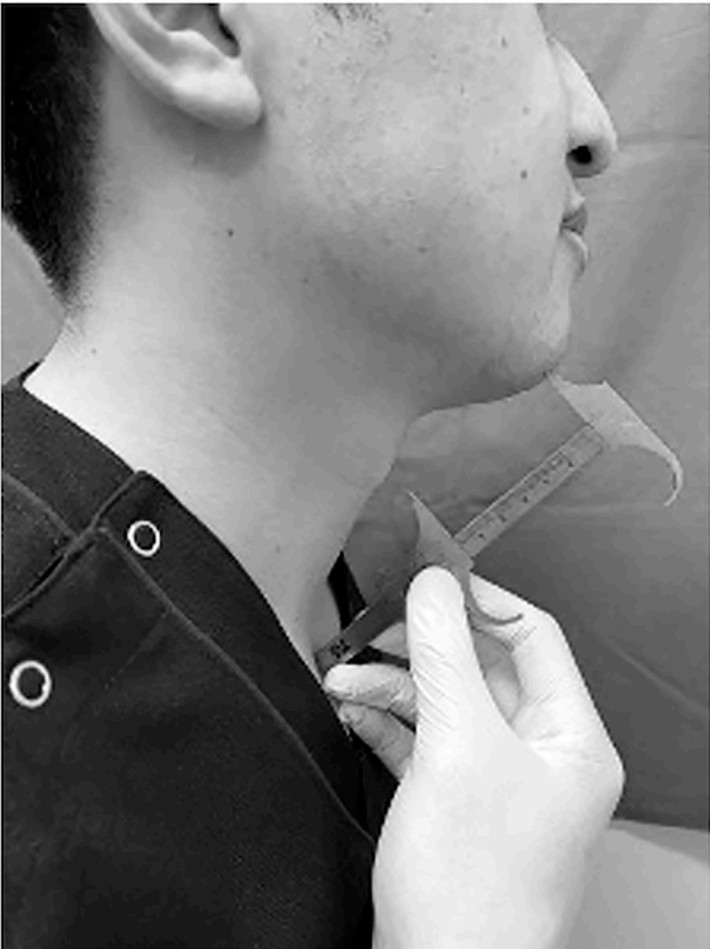

**Fig 2. Measuring method of the distance between the menton and laryngeal prominence.**

Whitney U-test were used for parametric and non-parametric data, respectively. Chi-square tests were used to compare the categorical data. Spearman's rank correlation coefficient was used to investigate the correlation between the Mallampati classification and distance from the menton to laryngeal prominence. Logistic regression analyses were conducted to investigate the influence of choking on the coughing reflex from food or drink and medical history of pneumonia. The variables used were age, sex, height, BMI, ECOG PS, RSST, modified Mallampati classification, and distance from the menton to the laryngeal prominence. Statistical significance was set at a p-value of $<0.05$.

## Ethical approval

This study was approved by the Ethics Committee of Shinshu University School of Medicine (approval code: 5127). The study protocol adhered to the ethical guidelines of the Declaration of Helsinki and the ethical guidelines for medical and health research involving humans by the Ministry of Health, Labor, and Welfare of Japan. The study details were explained to the study participants. Written consent was obtained from all the participants.

## Results

### Characteristics of studied participants

The median age of the participants (men: 92, women: 99) was 79.3 (range, 65–98) years. In total, 152 (79.6%) participants were evaluated for saliva swallowing tests more than twice. In the questionnaires regarding medical history, 22 participants (11.5%) answered that they had pneumonia in the previous year. Furthermore, 44 participants (23.0%) answered that they had "choking or coughing reflex from food or drinking" in the previous month. The patients were classified according to the modified Mallampati classifications as follows: class I, 23 patients (12.0%); class II, 45 patients (23.6%); class III, 69 patients (36.1%); and class IV, 54 patients (28.3%). The largest distance between the menton and laryngeal prominence in the 4-cm range (n = 50; 26.2%), followed by the 5-cm range (n = 44, 23.0%) and the 6-cm range (n = 42; 22.0%) (Table 1).

### Correlation between the modified Mallampati classification and the participants' characteristics and swallowing function

No significant correlations were found between the modified Mallampati classification and physical findings (age, sex, height, BMI, and PS). In addition, no significant relationship was observed between the modified Mallampati classification and the RSST (Mann–Whitney test, p = 0.311) (Table 2).

### Correlation between hyoid bone position and participants' characteristics and swallowing function

There were statistically significant correlations between the distance from the menton to the laryngeal prominence and the participants' age or sex (Table 3). The distance was longer in the participants younger than 75 years than those who were 75 years or older (Mann–Whitney test, p = 0.009). The distance was also longer in male than in female participants (Mann–Whitney U-test, p = 0.027). No significant associations were found with other physical findings. No significant association was found between the distance and RSST (Mann–Whitney test, p = 0.846).

### Relationship between the modified Mallampati classification and hyoid bone position

The correlation between the modified Mallampati classification and the distance between the menton and laryngeal prominence is shown in Fig 3. No significant correlation was observed between these variables (Spearman's rank correlation coefficient r = 0.023; p = 0.751).

### Correlation between either the oropharyngeal morphology or hyoid bone position and signs of pulmonary aspiration

We performed univariate and multivariate analyses to analyze the impact of the modified Mallampati classification and hyoid position on aspiration pneumonia. The objective variables were signs of aspiration (history of choking or cough reflex in the previous month and medical history of pneumonia in the previous year), and the explanatory variables were age, sex, height, BMI, ECOG PS, RSST, modified Mallampati classification, and distance between the menton and laryngeal prominence.

The results of the analyses of the factors that correlated with the experience of choking or cough reflex in the previous month are shown in Table 4. In the univariate analyses, there was

**Table 1. Characteristics of the study participants.**

|  | Number (%), Average (range) |
|---|---|
| **Age** |  |
| Average | 79.3 years old (65–98) |
| **Sex** |  |
| Male | 92 (48.2) |
| Female | 99 (51.8) |
| **Height** |  |
| Average | 157.2 cm (124.0–176.3) |
| **Weight** |  |
| Average | 55.3 kg (30–96) |
| **Body Mass Index (BMI)** |  |
| Average | 22.3 (14.1–34.5) |
| **ECOG performance status (PS)** |  |
| 0 | 103 (53.9) |
| 1 | 41 (21.5) |
| 2 | 23 (12.0) |
| 3 | 24 (12.6) |
| **Repeated saliva swallowing test (RSST)** |  |
| 0 times | 4 (2.1) |
| 1 times | 8 (4.2) |
| 2 times | 27 (14.1) |
| ≧ 3 times | 152 (79.6) |
| **Choking or cough reflex from food or drinking in the previous month** |  |
| Present | 44 (23.0) |
| Absent | 147 (77.0) |
| **Medical history of pneumonia in previous year** |  |
| Present | 22 (11.5) |
| Absent | 169 (88.5) |
| **Modified Mallampati classification** |  |
| I | 23 (12.0) |
| II | 45 (23.6) |
| III | 69 (36.1) |
| IV | 54 (28.3) |
| **Distance from the menton to the laryngeal prominence** |  |
| < 3.0 cm | 6 (3.1) |
| 3.0–3.9 cm | 18 (9.4) |
| 4.0–4.9 cm | 50 (26.2) |
| 5.0–5.9 cm | 44 (23.0) |
| 6.0–6.9 cm | 42 (22.0) |
| 7.0–7.9 cm | 23 (12.0) |
| ≧ 8.0 cm | 8 (4.2) |

ECOG: Eastern Cooperative Oncology Group

a statistically significant association between the experience of choking or cough reflex and BMI (Student's *t*-test, p = 0.019), Mallampati classification (Mann–Whitney U-test, p = 0.045), and hyoid position (Mann–Whitney U-test, p <0.001). The logistic regression analysis revealed that hyoid position was the only independently associated factor. The shorter the distance between the menton and laryngeal prominence, the greater the observed aspiration.

**Table 2. Correlation between the modified Mallampati classification and participants' characteristics.**

| | Modified Mallampati classification | | | | |
|---|---|---|---|---|---|
| | **I** | **II** | **III** | **IV** | |
| | **(n = 23)** | **(n = 45)** | **(n = 69)** | **(n = 54)** | **p-value** |
| **Age** | | | | | 0.645 |
| < 75 years old (n = 71) | 12 | 14 | 25 | 20 | |
| ≧ 75 years old (n = 120) | 11 | 31 | 44 | 34 | |
| **Sex** | | | | | 0.625 |
| Male (n = 92) | 11 | 26 | 28 | 27 | |
| Female (n = 99) | 12 | 19 | 41 | 27 | |
| **Height** | | | | | 0.658 |
| < 157 cm (n = 93) | 11 | 18 | 39 | 25 | |
| ≧ 157 cm (n = 98) | 12 | 27 | 30 | 29 | |
| **BMI** | | | | | 0.195 |
| < 25 (n = 143) | 20 | 36 | 47 | 40 | |
| ≧ 25 (n = 48) | 3 | 9 | 22 | 14 | |
| **ECOG Performance Status** | | | | | 0.829 |
| 0 (n = 103) | 11 | 25 | 38 | 29 | |
| 1–3 (n = 88) | 12 | 20 | 31 | 25 | |
| **Repeated saliva swallowing test (RSST)** | | | | | 0.311 |
| < 3 times (n = 39) | 6 | 8 | 18 | 7 | |
| ≧ 3 times (n = 152) | 17 | 37 | 51 | 47 | |

Mann–Whitney U test, BMI: body mass index, ECOG: Eastern Cooperative Oncology Group

In the univariate analyses, there was a statistically significant association between a medical history of pneumonia and age (Student's $t$-test, p <0.001), sex (Chi-square test, p = 0.046), BMI (Student's $t$-test, p < 0.001), PS (Mann–Whitney U-test, p <0.001), RSST (Chi-square test, p < 0.001), and modified Mallampati classification (Mann–Whitney U-test, p = 0.009). The multivariate logistic regression analysis revealed that age (p = 0.031), BMI (p = 0.045), and modified Mallampati classification (p = 0.045) were independently associated with a medical history of pneumonia. The higher the age, lower the BMI, and lower the modified Mallampati classification, the higher the incidence of aspiration pneumonia (Table 5).

## Discussion

The results of this study showed that the modified Mallampati classification significantly correlated with a medical history of aspiration pneumonia. Lower classification scores (i.e., more visible soft palate, uvula, fauces, and pillars) were associated with higher incidence of aspiration pneumonia (Fig 4). The incidence rates of pneumonia were 26.1%, 15.6%, 8.7%, and 5.6% in classes I, II, III, and IV, respectively. The adjusted odds ratio was 1.7 (95% confidence interval: 1.21–2.33), indicating that the patients were 1.8 times more likely to develop pneumonia for every one drop in class. The Mallampati classification evaluates the size of the tongue base relative to that of the oropharyngeal cavity [9]. The tongue is composed of intrinsic and extrinsic lingual muscles, and its position is determined by muscles, the hyoid bone, and the mandible, to which some lingula muscles are attached. Therefore, the size and position of the tongue base are likely to be influenced by lingual muscle volume, obesity, hyoid bone position, and suprahyoid muscle function. The Mallampati classification has been reported to be associated with obesity [11, 12]. However, in this study, no significant association was observed between

**Table 3. Correlation between the distance from the menton to the laryngeal prominence and participants' characteristics.**

| | Distance from the menton to the laryngeal prominence (cm) | | | | | | | |
| --- | --- | --- | --- | --- | --- | --- | --- | --- |
| | < 3.0 | 3.0–3.9 | 4.0–4.9 | 5.0–5.9 | 6.0–6.9 | 7.0–7.9 | ≧ 8.0 | p-value |
| | (n = 6) | (n = 18) | (n = 50) | (n = 44) | (n = 42) | (n = 23) | (n = 8) | |
| **Age** | | | | | | | | **0.009** |
| < 75 years old (n = 71) | 1 | 3 | 19 | 13 | 17 | 15 | 3 | |
| ≧ 75 years old (n = 120) | 5 | 15 | 31 | 31 | 25 | 8 | 5 | |
| **Sex** | | | | | | | | **0.027** |
| Male (n = 92) | 0 | 7 | 24 | 23 | 15 | 17 | 6 | |
| Female (n = 99) | 6 | 11 | 26 | 21 | 27 | 6 | 2 | |
| **Height** | | | | | | | | 0.322 |
| < 157 cm (n = 93) | 5 | 10 | 22 | 21 | 24 | 9 | 2 | |
| ≧ 157 cm (n = 98) | 1 | 8 | 28 | 23 | 18 | 14 | 6 | |
| **BMI** | | | | | | | | 0.693 |
| < 25 (n = 143) | 6 | 12 | 40 | 31 | 30 | 17 | 7 | |
| ≧ 25 (n = 48) | 0 | 6 | 10 | 13 | 12 | 6 | 1 | |
| **ECOG Performance Status** | | | | | | | | 0.444 |
| 0 (n = 103) | 2 | 8 | 28 | 25 | 22 | 13 | 5 | |
| 1–3 (n = 88) | 4 | 10 | 22 | 19 | 20 | 10 | 3 | |
| **Repeated saliva swallowing test (RSST)** | | | | | | | | 0.846 |
| < 3 (n = 39) | 3 | 2 | 10 | 9 | 7 | 5 | 3 | |
| ≧ 3 (n = 152) | 3 | 16 | 40 | 35 | 35 | 18 | 5 | |

Mann–Whitney U test

BMI: body mass index, ECOG: Eastern Cooperative Oncology Group

the modified Mallampati class and BMI. This study included adults aged 65 years and older; therefore, there might be no association between obesity and Mallampati classification in individuals older than 65 years. In addition, the modified Mallampati classification was not associated with age or sex, which may indicate the functional and morphological characteristics of the tongue and suprahyoid musculature.

Additionally, the position of the hyoid bone was evaluated as a predictor of swallowing function and aspiration pneumonia. The position of the hyoid bone, which is determined by the suprahyoid muscles attached to the skull or mandible, can be altered by changes in the size and function of the suprahyoid muscles, consequently affecting swallowing function. Previous studies have reported that hyoid bone position is associated with age and sex [10, 13]. The results of this study showed a significant association between hyoid bone position and either age or sex; participants younger than 75 years had a lower hyoid bone position (longer distance between the menton and laryngeal prominence) than that in participants older than 75 years, and men had a lower hyoid bone position than that in women. Moreover, the present study findings did not reveal a relationship between hyoid bone position and height. These results suggest that hyoid position is influenced by sex and age. Swallowing movements require elevation of the hyoid bone. Men have greater muscle strength than women at younger ages [14–16]. The low position of the hyoid bone in men and younger individuals may be related to the muscle strength required to elevate it.

The relationship between the modified Mallampati classification and the position of the hyoid bone was investigated because the size and position of the base of the tongue (modified Mallampati classification) are believed to be influenced by the position of the hyoid bone. However, the results of this study showed no significant correlations. These results suggested

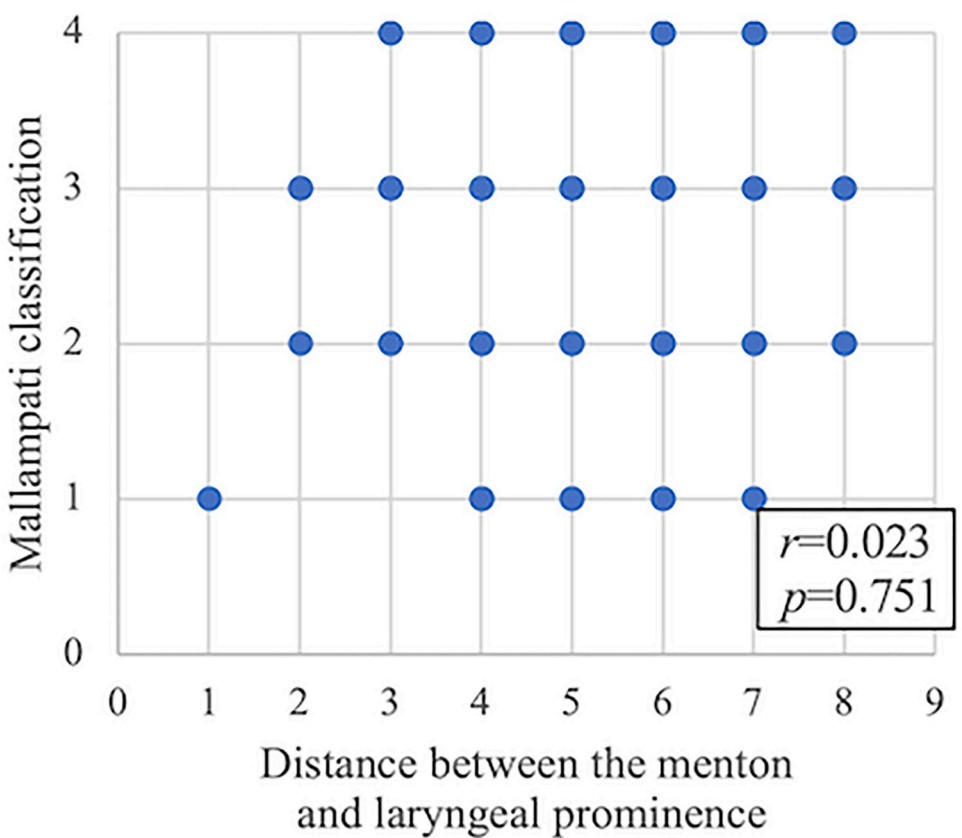

**Fig 3. Correlation between the modified Mallampati classification and the distance between the menton and laryngeal prominence.**

that the decrease in the size and position of the base of the tongue was attributed to factors other than the position of the lower hyoid bone.

In this study, the modified Mallampati classification was significantly associated with a medical history of pneumonia, but not with the experience of choking or cough reflex from food or drinking. However, there may be some inconsistencies in the results. Although two indices were used in this study, there was a difference in what was assessed between the two. The experience of choking or cough reflex from food or drinking inquired about the experience during swallowing exercise (during eating or drinking), that is, the results in the dynamic state, whereas a medical history of pneumonia is an assessment that includes the results in the non-exercising (i.e., static) state. Aspiration pneumonia may be associated with silent unobserved aspirations [17–19]. In the present study, we used the RSST to screen for dysphagia. In the RSST, the patient is instructed to swallow their own saliva as many times as possible in 30 seconds. The number of swallows completed successfully was counted by palpating the patient's laryngeal movements. Although many factors, such as pharyngeal movement, oral function, salivation, and cognition, are believed to influence the ability to perform the RSST [20], the muscular function associated with swallowing (swallowing exercise) may be the most important factor. In the current study, we examined the association between the modified Mallampati classification and the RSST scores; no significant association was observed. The Mallampati classification does not assess the swallowing performance. When evaluating aspiration, retention of saliva, food, and drinks in the oral cavity is important. The volume of the

**Table 4. Univariate and multivariate analyses of the factors which correlate with history of choking or cough reflex from food or drinking in the previous month.**

| | | Results of univariate analysis | | | Results of multivariate logistic analysis | | |
|---|---|---|---|---|---|---|---|
| | | Experience of choking or cough reflex in the previous month | | p-value | Parameter estimation value | Standard error | p-value |
| | | Yes (n = 44) | No (n = 147) | | | | |
| Age (mean ± SE, years) | | 80.9±1.3 | 78.8±0.7 | 0.163* | -0.013 | 0.030 | 0.659 |
| Sex | | | | 0.335† | -0.428 | 0.273 | 0.116 |
| Male | 92 | 24 | 68 | | | | |
| Female | 99 | 20 | 79 | | | | |
| Height (mean ± SE, cm) | | 157.1±1.5 | 157.2±0.8 | 0.951* | -0.005 | 0.030 | 0.866 |
| BMI | | 21.1±0.6 | 22.7±0.3 | 0.019* | -0.06 | 0.053 | 0.256 |
| ECOG Performance Status | | | | 0.094§ | 0.150 | 0.217 | 0.491 |
| 0 | 103 | 20 | 83 | | | | |
| 1 | 41 | 8 | 33 | | | | |
| 2 | 23 | 8 | 15 | | | | |
| 3 | 24 | 8 | 16 | | | | |
| Repeated saliva swallowing test (RSST) | | | | 0.087† | 0.316 | 0.246 | 0.199 |
| < 3 times | 39 | 13 | 26 | | | | |
| ≧ 3 times | 152 | 31 | 121 | | | | |
| Modified Mallampati classification | | | | 0.045§ | -0.301 | 0.197 | 0.127 |
| I | 23 | 8 | 15 | | | | |
| II | 45 | 11 | 34 | | | | |
| III | 69 | 18 | 51 | | | | |
| IV | 54 | 7 | 47 | | | | |
| Distance from the menton to the laryngeal prominence | | | | <0.001§ | -0.662 | 0.158 | <0.001 |
| < 3.0 cm | 6 | 5 | 1 | | | | |
| 3.0–3.9 cm | 18 | 7 | 11 | | | | |
| 4.0–4.9 cm | 50 | 15 | 35 | | | | |
| 5.0–5.9 cm | 44 | 12 | 32 | | | | |
| 6.0–6.9 cm | 42 | 1 | 41 | | | | |
| 7.0–7.9 cm | 23 | 3 | 20 | | | | |
| ≧ 8.0 cm | 8 | 1 | 7 | | | | |

\*: Student's *t*-test

†: Chi-square test

§: Mann–Whitney U-test

BMI: body mass index, ECOG: Eastern Cooperative Oncology Group

tongue root plays an important role in the blockade among the oral cavity, nasal cavity, and pharynx. Given that participants with Mallampati class I (small volume of the base of the tongue relative to the oropharyngeal cavity) had a higher incidence of pneumonia suggests that the reduced ability of the tongue base to block the path between the oral cavity and the pharynx may have influenced the development of aspiration pneumonia.

In the present study, the results showed that the higher the position of the hyoid bone, the more likely it was to cause choking and cough reflex during eating and drinking. In general, it is believed that in older adults, the suprahyoid muscle group relaxes, the position of the hyoid bone decreases, and the force to pull up the hyoid bone weakens, resulting in decreased swallowing function and increased susceptibility to aspiration, often accompanied by other complications [3, 6, 10, 21, 22]; however, the results of the present study contradict this hypothesis.

**Table 5. Univariate and multivariate analyses of the factors correlated to medical history of pneumonia in the previous year.**

| | | Results of univariate analysis | | | Results of multivariate logistic analysis | | |
|---|---|---|---|---|---|---|---|
| | | Medical history of the pneumonia in previous year | | p-value | Parameter estimation value | Standard error | p-value |
| | | Yes (n = 22) | No (n = 169) | | | | |
| Age (mean ± SE, years) | | 86.5±1.8 | 78.3±0.7 | < 0.001* | 0.092 | 0.042 | 0.031 |
| Sex | | | | 0.046† | -0.464 | 0.385 | 0.229 |
| Male | 92 | 15 | 77 | | | | |
| Female | 99 | 7 | 92 | | | | |
| Height (mean ± SE cm) | | 158.6±2.2 | 156.9±0.8 | 0.467* | 0.033 | 0.043 | 0.432 |
| BMI | | 19.1±0.8 | 22.7±0.3 | <0.001* | -0.186 | 0.092 | 0.045 |
| ECOG Performance Status | | | | <0.001§ | 0.259 | 0.270 | 0.339 |
| 0 | 103 | 6 | 97 | | | | |
| 1 | 41 | 3 | 38 | | | | |
| 2 | 23 | 4 | 19 | | | | |
| 3 | 24 | 9 | 15 | | | | |
| Repeated saliva swallowing test (RSST) | | | | < 0.001† | 0.499 | 0.309 | 0.106 |
| < 3 times | 39 | 11 | 28 | | | | |
| ≧ 3 times | 152 | 11 | 141 | | | | |
| Modified Mallampati classification | | | | 0.009§ | -0.573 | 0.287 | 0.045 |
| I | 23 | 6 | 17 | | | | |
| II | 45 | 7 | 38 | | | | |
| III | 69 | 6 | 63 | | | | |
| IV | 54 | 3 | 51 | | | | |
| Distance from the menton to the laryngeal prominence | | | | 0.890§ | 0.116 | 0.200 | 0.563 |
| < 3.0 cm | 6 | 2 | 4 | | | | |
| 3.0–3.9 cm | 18 | 3 | 15 | | | | |
| 4.0–4.9 cm | 50 | 3 | 47 | | | | |
| 5.0–5.9 cm | 44 | 4 | 40 | | | | |
| 6.0–6.9 cm | 42 | 6 | 36 | | | | |
| 7.0–7.9 cm | 23 | 2 | 21 | | | | |
| ≧ 8.0 cm | 8 | 2 | 6 | | | | |

*: Student's *t*-test

†: Chi-square test

§: Mann-Whitney U-test

SE: standard error, BMI: body mass index, ECOG: Eastern Cooperative Oncology Group

As mentioned above, the position of the hyoid bone appears to be related to suprahyoid muscle strength. In addition, there was no association between hyoid bone position and the RSST in the present study. Accordingly, we speculate that the dysfunction of the suprahyoid muscles, which is believed to be a factor in the decline in swallowing performance, is not directly related to the drop in hyoid bone position.

The strength of the present study is that it focuses on Mallampati classification that is easy to measure on a daily clinical practice, in addition, examines the association between it and pneumonia, which has large number of patients. This study has some limitations, including the small number of participants. Further, history of pneumonia was confirmed by interview, which has possibility to be influenced by recall bias due to self-reporting. Additionally, the reproducibility and reliability of the Mallampati classification were not evaluated. In addition,

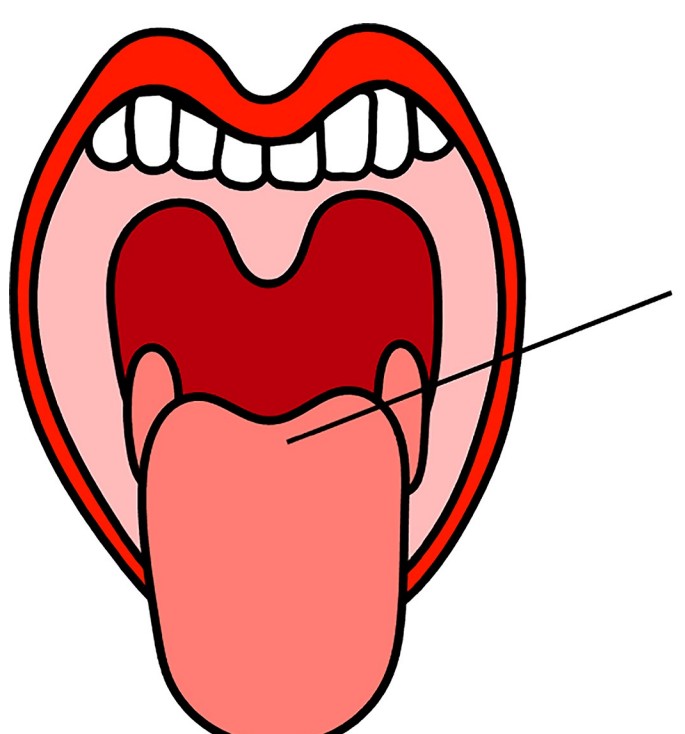

Lower Mallampati classification (smaller size of the base of the tongue in the oropharyngeal cavity)

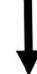

Higher incidence of pneumonia

**Fig 4. Lower classification scores were associated with higher incidence of pneumonia.**

we did not take into account the presence of mandibular hypoplasia or different tongue sizes individually.

As integration into routine clinical practice, various medical professionals can use the Mallampati classification method in medical check-ups for older adults. For the clinical application, we will further investigate the usefulness of the Mallampati classification in predicting pneumonia in older patients.

## Conclusion

The absence of an association between the Mallampati classification and the RSST, which assesses swallowing function, and the lack of a strong association between the Mallampati classification and dysphagia suggest that this classification does not predict the occurrence of aspiration pneumonia, but rather assesses community-acquired pneumonia as a whole.

## Acknowledgments

We are grateful to Drs. T. Koike, S. Uehara, A. Takizawa, A. Shibata, S. Nagashio, Y. Nakanishi, A. Umehara, H. Aizawa, Y. Koyama, D. Akita, and Y. Kusafuka for their cooperation in this study.

We would like to thank Editage (www.editage.jp) for English language editing.

## Author Contributions

**Conceptualization:** Eiji Kondo.

**Data curation:** Hironobu Fukuda.

**Formal analysis:** Hironobu Fukuda.

**Investigation:** Hironobu Fukuda, Yuki Sakai.

**Methodology:** Hironori Sakai.

**Supervision:** Hiroshi Kurita.

**Writing – original draft:** Jianping Liu, Hironobu Fukuda.

**Writing – review & editing:** Hiroshi Kurita.

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
