## [Decision Letter · Decision Letter 0]

9 Jan 2024

PONE-D-23-40581Oropharyngeal morphology (modified Mallampati classification) could be a useful screening tool for the risk of pneumonia in the older adultsPLOS ONE

Dear Dr. FUKUDA,

Thank you for submitting your manuscript to PLOS ONE. After careful consideration, we feel that it has merit but does not fully meet PLOS ONE’s publication criteria as it currently stands. Therefore, we invite you to submit a revised version of the manuscript that addresses the points raised during the review process.

We look forward to receiving your revised manuscript.

Kind regards,

Fentahun Adane Nigat, MSc., PhD

Academic Editor

PLOS ONE

Journal Requirements:

2. In the online submission form, you indicated that [Insert text from online submission form here]. 

Additional Editor Comments:

Dear Authors,

I have reviewed the manuscript titled "Oropharyngeal morphology (modified Mallampati classification) could be a useful screening tool for the risk of pneumonia in the older adults" (Manuscript Number: PONE-D-23-40581) and have some questions and comments for the authors.

General basic Questions:

1. What specific criteria were used in the modified Mallampati classification to assess oropharyngeal morphology?

2. How was dysphagia assessed using the repetitive saliva swallowing test (RSST)? Can you provide more details about the RSST?

3. Were there any specific inclusion or exclusion criteria for the recruitment of the 191 older participants?

4. What statistical methods were employed to analyze the data and determine the correlation between the modified Mallampati classification and aspiration pneumonia?

5. Were there any limitations or potential confounding factors in the study design that could have influenced the results?

Section specific questions:

6. Abstract: Could you provide more information about the specific methods used to assess oropharyngeal morphology and hyoid bone position? Additionally, please clarify the criteria used to define dysphagia and aspiration pneumonia.

7. Results: The correlation between the modified Mallampati classification and a medical history of aspiration pneumonia is mentioned. However, it would be helpful to include the strength of this correlation (e.g., correlation coefficient or p-value) to better understand the relationship.

8. Results: The lack of significant association between the Mallampati classification and the repetitive saliva swallowing test (RSST) or history of choking/gag reflex is intriguing. Are there any potential explanations for this finding? If so, please discuss them.

9. Discussion: It would be valuable to provide a more detailed discussion on the clinical implications of using the modified Mallampati classification as a screening tool for aspiration pneumonia. How might this classification be integrated into routine clinical practice? Are there any limitations or potential challenges in implementing this screening method?

10. Funding Disclosure: The manuscript states that the authors received no specific funding for this work. However, it would be helpful to clarify whether there were any sources of funding that indirectly supported the study or the authors' involvement.

11. Competing Interests: The authors have declared no competing interests. It would be beneficial to include a statement explaining that none of the authors have any financial or non-financial relationships that could be perceived as potentially influencing the interpretation of the results or biasing the study.

12. Ethics Statement: The manuscript mentions that the research was approved by the Ethics Committee. Please provide more details regarding the specific ethics approval, such as the name of the committee, the approval number, and any relevant ethical considerations that were addressed.

13. General: Please consider providing additional information regarding the demographic characteristics of the study participants, such as age range, gender distribution, and any relevant comorbidities. This information would help readers better understand the generalizability of the study findings.

14. Language and Clarity: Some sections of the manuscript could be further clarified to improve readability and understanding. I recommend a thorough proofreading to ensure accurate grammar, sentence structure, and scientific terminology throughout the manuscript.

15. Figures and Tables: Are there any figures or tables included in the manuscript to visually support the results or enhance the understanding of the study? If not, I suggest considering the inclusion of relevant visual aids to complement the text.

16. Please address these questions and comments in your revised manuscript.

Comments:

1. The abstract provides a clear overview of the study objectives and methods.

2. The results indicating a significant correlation between the modified Mallampati classification and a medical history of aspiration pneumonia are interesting. However, it would be helpful to discuss the clinical implications of this finding in more detail.

3. It would be beneficial to include a discussion on the potential applications of the modified Mallampati classification as a screening tool for dysphagia and aspiration pneumonia in clinical practice.

4. Providing additional information on the demographic characteristics of the study participants, such as age distribution and gender ratio, would enhance the understanding of the study population.

5. Consider discussing the limitations of using self-reported history of choking or coughing reflex as a measure of aspiration signs, as this may introduce recall bias.

Reviewers' comments:

Reviewer's Responses to Questions

**Comments to the Author**

1. Is the manuscript technically sound, and do the data support the conclusions?

Reviewer #1: Yes

Reviewer #2: Yes

2. Has the statistical analysis been performed appropriately and rigorously? 

Reviewer #1: Yes

Reviewer #2: Yes

3. Have the authors made all data underlying the findings in their manuscript fully available?

Reviewer #1: Yes

Reviewer #2: Yes

4. Is the manuscript presented in an intelligible fashion and written in standard English?

Reviewer #1: No

Reviewer #2: Yes

5. Review Comments to the Author

Reviewer #1: I would like to express sincere thanks for the study you have done. Your contribution to evidence-based public health intervention is substantial, particularly in the context of the elderly population. However, I would like to request a major revision to your work.

Reviewer #2: Dear Editorial Team,

- I sincerely appreciate you providing me the opportunity to review this significant scientific paper.

- I express my gratitude to the authors for their excellent work in examining the risk of pneumonia through non-invasive procedures, considering modifications in the morphology of the oropharynx and the position of the hyoid bone.

General comments

- The work could improve clinical practices and is crucial for the management of elderly patients.

- The writing is excellent, and the arrangement follows scientific guidelines.

Specific comments

1. Title: The title is informative and lucid.

2. Abstract: nicely put. However, the sampling strategy is left out. Add the sampling technique now.

3. Introduction: The writing is well done

4. Methods

- According to the authors, all research-related surveys were completed between May 16, 2021, and June 15, 2022. What makes the addition of one month (May 16, 2022 to June 15, 2022) noteworthy? The study took place between May 16, 2021 and May 15, 2022, if it was to last a full year.

- The study included 191 participants in total. What technique was used for sampling? The information was gathered at eleven hospitals. What percentage of participants was each hospital expected to contribute? Or was it just that any number of recruited subjects was employed in the research?

- The study included people who were 65 years of age or older as participants. Why should you use people who are 65 years of age or older? Why not sixty or fifty? or somebody else?

- Addition of dependent and independent variables was anticipated by the authors.

- Addition of operational definitions is necessary.

5. Results

- Well written

6. Discussion

- The findings of the study are well discussed with other studies.

6. PLOS authors have the option to publish the peer review history of their article (what does this mean?). If published, this will include your full peer review and any attached files.

Reviewer #1: **Yes: **Bickes Wube Sume

Reviewer #2: **Yes: **Hussen Abdu

---

## [Author Response · Author response to Decision Letter 0]

9 Feb 2024

We thank you and the reviewers for your thoughtful suggestions and insights. The manuscript has benefited from these insightful suggestions. I look forward to working with you and the reviewers to move this manuscript closer to publication in PLoS ONE.

The manuscript has been rechecked and the necessary changes have been made in accordance with the reviewers’ suggestions. The responses to all comments have been prepared and attached herewith. 

Thank you for your consideration. I look forward to hearing from you.

---

## [Decision Letter · Decision Letter 1]

3 Apr 2024

Screening of Aspiration Pneumonia using the Modified Mallampati Classification Tool in Older Adults

PONE-D-23-40581R1

Dear Dr. FUKUDA,

We’re pleased to inform you that your manuscript has been judged scientifically suitable for publication and will be formally accepted for publication once it meets all outstanding technical requirements.

Kind regards,

Sethu Thakachy Subha, M.S

Academic Editor

PLOS ONE

Additional Editor Comments (optional):

Reviewers' comments:

Reviewer's Responses to Questions

**Comments to the Author**

1. If the authors have adequately addressed your comments raised in a previous round of review and you feel that this manuscript is now acceptable for publication, you may indicate that here to bypass the “Comments to the Author” section, enter your conflict of interest statement in the “Confidential to Editor” section, and submit your "Accept" recommendation.

Reviewer #1: All comments have been addressed

2. Is the manuscript technically sound, and do the data support the conclusions?

Reviewer #1: Yes

3. Has the statistical analysis been performed appropriately and rigorously? 

Reviewer #1: Yes

4. Have the authors made all data underlying the findings in their manuscript fully available?

Reviewer #1: Yes

5. Is the manuscript presented in an intelligible fashion and written in standard English?

Reviewer #1: Yes

6. Review Comments to the Author

Reviewer #1: Dear editor, all of my comments and questions were addressed. I think this manuscript is feasible for publication.

7. PLOS authors have the option to publish the peer review history of their article (what does this mean?). If published, this will include your full peer review and any attached files.

Reviewer #1: **Yes: **Bickes Wube Sume
